# ABPP-HT*—Deep Meets Fast for Activity-Based Profiling of Deubiquitylating Enzymes Using Advanced DIA Mass Spectrometry Methods

**DOI:** 10.3390/ijms23063263

**Published:** 2022-03-17

**Authors:** Hannah B. L. Jones, Raphael Heilig, Simon Davis, Roman Fischer, Benedikt M. Kessler, Adán Pinto-Fernández

**Affiliations:** 1Target Discovery Institute, Centre for Medicines Discovery, Nuffield Department of Medicine, University of Oxford, Roosevelt Drive, Oxford OX3 7FZ, UK; hannah.jones@ndm.ox.ac.uk (H.B.L.J.); r.heilig@directbox.com (R.H.); simon.davis2@ndm.ox.ac.uk (S.D.); roman.fischer@ndm.ox.ac.uk (R.F.); 2Chinese Academy for Medical Sciences Oxford Institute, Nuffield Department of Medicine, University of Oxford, Roosevelt Drive, Oxford OX3 7FZ, UK

**Keywords:** activitomics, activity-based probes, chemical biology, data-independent acquisition mass spectrometry (DIA), deubiquitylating enzymes, drug discovery, mass spectrometry, proteomics, ubiquitin

## Abstract

Activity-based protein profiling (ABPP) uses a combination of activity-based chemical probes with mass spectrometry (MS) to selectively characterise a particular enzyme or enzyme class. ABPP has proven invaluable for profiling enzymatic inhibitors in drug discovery. When applied to cell extracts and cells, challenging the ABP-enzyme complex formation with a small molecule can simultaneously inform on potency, selectivity, reversibility/binding affinity, permeability, and stability. ABPP can also be applied to pharmacodynamic studies to inform on cellular target engagement within specific organs when applied to in vivo models. Recently, we established separate high depth and high throughput ABPP (ABPP-HT) protocols for the profiling of deubiquitylating enzymes (DUBs). However, the combination of the two, deep and fast, in one method has been elusive. To further increase the sensitivity of the current ABPP-HT workflow, we implemented state-of-the-art data-independent acquisition (DIA) and data-dependent acquisition (DDA) MS analysis tools. Hereby, we describe an improved methodology, ABPP-HT* (enhanced high-throughput-compatible activity-based protein profiling) that in combination with DIA MS methods, allowed for the consistent profiling of 35–40 DUBs and provided a reduced number of missing values, whilst maintaining a throughput of 100 samples per day.

## 1. Introduction

Activity-based probes (ABPs) react with the active site of an enzyme to inform on its activity. Typically, they are comprised of a warhead that binds irreversibly, a specificity motif to ensure selectivity for a particular enzyme or enzyme family, and a reporter tag [1,2,3]. Their specificity means that ABPs can be applied to monitor the activity of enzymes in a cellular environment without the need for enzyme purification. Depending on the ABP, they can be applied to intact cells, or used with cellular lysates directly [4,5]. ABPs are invaluable for profiling potential enzymatic inhibitors in the early stages of drug discovery [6,7]. The prevention of ABP-enzyme complex formation by a small molecule inhibitor can inform on multiple compound parameters such as potency and selectivity in lysates, reversibility/binding affinity, permeability and stability within intact cells. ABPs used in combination with pharmacodynamic studies can also inform on inhibitor target engagement within cells and specific organs in vivo.

A family of enzymes that are currently being targeted for therapeutic inhibition are deubiquitylating enzymes (DUBs). DUBs oppose the process of ubiquitination, a post-translational modification (PTM) of proteins responsible for the regulation of numerous cellular functions, such as degradation and signaling [8,9]. In some cases, the removal of ubiquitin from specific protein targets by a DUB can prevent proteasomal proteolysis or degradation by the autophagosome/lysosome system [10]. Currently, DUB inhibitors are being developed for therapeutic treatment of a number of diseases including Parkinson’s disease and cancer [11].

Previously, we have accomplished in-depth profiling of DUB inhibitors using a combination of ABP immunoprecipitation and LC-MS/MS proteomic analysis, a methodology known as ABPP (activity-based protein profiling) [12]. With further improvements of this technique, we have demonstrated the ability of a ubiquitin-based ABP to form complexes with 74 endogenous DUBs in MCF-7 breast cancer cells [13]. These 74 DUBs represented the majority of active cysteine-protease DUBs, including distinguishable isoforms, expressed in the human genome [13,14]. The Ub-based ABP used in this study is comprised of a HA tag for immunoprecipitation, ubiquitin for specificity, and a propargylamine warhead for binding to the active site cysteine (HA-Ub-PA).

This in-depth profiling was achieved with low pH C18 HPLC pre-fractionation, resulting in 10 samples concatenated from 60 fractions that were subsequently analysed using a 60 min gradient per fraction on an Orbitrap LC-MS/MS system [13]. While this in-depth ABPP would be particularly valuable for a late-stage drug candidate, it does not offer the necessary throughput for compound screening in the early stages of drug discovery. Without the pre-fractionation of samples, a 60 min orbitrap LC-MS/MS gradient typically leads to the identification and quantitation of ~30–40 DUBs [15,16]. Whilst the reduced number of DUBs identified can indeed act as a representative panel of DUBs for drug screening, the throughput of around 10 samples per day is still limiting drug discovery applications.

To overcome this, we recently developed a methodology that employed the use of the Agilent Bravo AssayMAP liquid handling platform to improve the ABPP’s immunoprecipitation and sample preparation throughput, and applied this in combination with runtimes of 15 min per sample on an Evosep/Bruker timsTOF LC-MS/MS [17]. This methodology, termed ABPP-HT, enabled a 10-fold increase in throughput, allowing for the analysis of 100 samples per day. However, the increase in throughput led to a reduced profiling depth, typically resulting in the identification and quantitation of ~15–25 DUBs. Although these DUBs can still be used as a representative panel for screening purposes, the decreased DUBome depth makes the methodology less suitable for low abundance DUBs and may mean key compound cross-reactivity data is overlooked.

In this work, we sought to improve the DUBome profiling depth whilst retaining the increased throughput offered by the Bravo/Evosep/timsTOF combination. This was achieved via the exploration of improved data acquisition and analysis methodology. Our previous methodology employed a widely used search engine software (MaxQuant) in combination with data-dependent acquisition (DDA) where the most abundant precursor ions are selected for further fragmentation. While powerful, this semi-stochastic precursor picking leads to missing values for low abundance peptides, resulting in data reproducibility issues and reduced sensitivity [18]. Data-independent acquisition (DIA) is advantageous as it selects all peptides within a given *m*/*z* window, giving reproducible precursor ions, leading to fewer missing values. Due to the trapped ion mobility capabilities of the timsTOF, the DIA data collection method, diaPASEF (parallel accumulation-serial fragmentation combined with data-independent acquisition) provides the opportunity to sample all peptide precursor ions, resulting in improved depth and reproducibility [19].

Recently there has been an increase in the availability of freely available proteomic software packages with DIA analysis capabilities, which work with *in silico* libraries, negating the time and cost associated with project specific libraries. Those that are free and are compatible with the diaPASEF data format include DIA-NN [20,21], and Maxquant’s MaxDIA [22]. Here, for data collected using the ABPP-HT methodology, we performed an in-depth comparison of free proteomic software packages including Maxquant and Fragpipe [23,24,25,26] for DDA analysis and DIA-NN and maxDIA for DIA analysis [20,22] (Figure 1). By applying DUB inhibitors with established potency/selectivity features we were able to examine the reproducibility, sensitivity and accuracy of each data acquisition and analysis method. Through this comparison, we found that the DUBome profiling depth of the ABPP-HT methodology could be improved to match that of the lower throughput methodology, in particular when combined with DIA/DIA-NN, whilst the data reproducibility and quantitative accuracy is maintained. We termed this methodology, enhanced high-throughput-compatible activity-based protein profiling (ABPP-HT*).

## 2. Results

### 2.1. ABPP-HT DUBome Depth Search Engine Comparison

With samples processed using the ABPP-HT methodology, differences in DUBome depth were evaluated with different proteomic software packages using data acquired by both DDA and DIA. In the case of DIA-acquired data, the library-free mode was applied to maintain the high-throughput nature of the experimentation whilst reducing costs. Here, we examined data obtained using both with and without “Match-between-runs” (MBR), a feature that helps to minimise the intrinsic missing value issue in proteomics by identifying peptides by ID transfer with tandem mass spectra from aligned runs, based on the *m*/*z*, charge state, retention time, and ion mobility (if available). We felt it important to evaluate data both with and without this feature as the size of our sample cohort is a variable that may alter the efficiency of MBR. It is of note that DIA-NN recommends the match-between-runs (MBR) setting where the library-free mode is being applied, and this should be considered when interpreting this dataset. The samples used for these comparisons were comprised of MCF-7 cell lysates that were probe-labelled and immunoprecipitated by the HA tag of the HA-Ub-PA probe in triplicate using the high-throughput method of combining Agilent’s liquid handling platform, Evosep liquid chromatography and a timsTOF mass spectrometer as outlined previously [17].

Figure 1A is a direct comparison of the total number of DUBs identified by DIA-NN and maxDIA for DIA data, as well as Fragpipe and Maxquant for DDA-acquired data. In all cases, other than maxDIA, applying match-between-runs (MBR) increased the identification and quantification of DUBs by LC-MS/MS, where samples from immunoprecipitations analysed in triplicate were matched. However, MBR only resulted in the identification of 1–3 extra DUBs on average, demonstrating the tractability of the methodology regardless of sample number. For analysis with Fragpipe, we compared the difference between 1 and 2 ion quantitation. This quantitation by Fragpipe’s ionquant represents the number of quantifiable ions required for protein quantification. Previously, 1 ion quantitation has been shown to have a comparable median CV (coefficient of variation) to Maxquant’s 2 peptide minimum quantitation [26]. The increased stringency of a 2 ion minimum for quantitation vs. a 1 ion minimum resulted in a reduction in DUBome depth. It is worth noting that for 1 ion minimum quantitation, with or without MBR, all identified DUBs had a CV of ≤ 20%, suggesting that the increased depth attained with this reduced stringency does not negatively impact the quantitative accuracy of the identified DUBome. From this dataset, we found that the application of both Maxquant in DDA mode and maxDIA in DIA mode, resulted in a significantly reduced DUBome depth, with the number of reproducibly identified DUBs (CV ≤ 0.2) reduced by approximately half as compared to those identified by DIA-NN, and approximately a third compared to those identified by Fragpipe. From this initial result, we ascertained that both Maxquant and maxDIA were not the optimal tools for the analysis of ABPP-HT data, and subsequent comparisons were performed using Fragpipe and DIA-NN.

Figure 1A demonstrates that for the same samples subjected to different acquisition modes, DIA data analysed with DIA-NN is able to quantify more DUBs on average when compared to DDA data analysed with Fragpipe. All DUBs identified by DIA-NN are common to Fragpipe, with DIA-NN identifying an additional 10 DUBs that are not quantified by Fragpipe (Figure 1B). The high levels of commonality in DUBs identified by both Fragpipe and DIA-NN gives further confidence that the additional depth achieved by both methods, when compared to Maxquant and maxDIA, represents accurately identified DUBs. To ascertain whether the additional DUBs identified by both methods are a direct result of HA-Ub-PA enrichment, the HA-Ub-PA samples were compared to a no HA-Ub-PA control sample (Figure 1C,D). All DUBs quantified from the Fragpipe analysis (2 ion minimum quantitation with MBR) were enriched > 10-fold (other than USP4) with the majority only present in samples with HA-Ub-PA, meaning the identification of these DUBs is likely attributable to ABP binding and subsequent enrichment (Figure 1C). Contrastingly, ~50% of the DUBs quantified by DIA-NN (MBR) had valid intensity values in samples both with and without HA-Ub-PA. However, and reassuringly, all DUBs, except for USP15, USP34, USP38, USP48 and OTUD7B, were enriched > 10-fold where HA-Ub-PA was present (Figure 1D). The increase in the number of DUBs present in both the no-probe and probe-labelled samples with DIA-NN analysis is likely attributable to the increased sensitivity associated with diaPASEF data when compared to DDA data [19], as demonstrated by the increase in the number of proteins quantified overall (total number of identified proteins: DIA-NN = 912, Fragpipe = 218). These additional proteins including the DUBs detected in the no probe sample come from co-immunoprecipitated complexes as well as non-specific background binding of proteins to the antibody and beads during the immunoprecipitation. However, it is important to note that in the presence of competitive HA-Ub-PA binding to the anti-HA antibody, these background proteins may not be present in the sample elution at all. Whilst a no ABP control is important to ascertain which DUBs could potentially bind as background, these DUBs should still be considered as cross-reactive to an inhibitor if they are immunoprecipitated in a concentration-dependent manner. Therefore, absolute quantitation of these DUBs for the identification of IC_50_ values should be treated with caution, and in some cases, normalised to a no-probe condition.

### 2.2. Reproducibility and Repeatability of Fragpipe and DIA-NN for ABPP-HT Data

To ensure that the additional DUBome depth attained with both Fragpipe and DIA-NN was reproducible, we used the same probe-labelled samples in triplicate to examine the number of unique peptides identified per DUB (Figure 2A), the coefficient of variation for the DUBome (Figure 2B), and the number of missing values within the DUBome (Figure 2C). The number of unique peptides identified per DUB followed the same trend regardless of whether Fragpipe (DDA) or DIA-NN (DIA) were used, or the settings applied, giving further confidence that the additional depth achieved by the data acquisition methods and software is not artifactual. The increased sensitivity of DIA data vs. DDA data is again clear, with DIA-NN consistently identifying more peptides than Fragpipe. DUBs that are identified by DIA-NN and not Fragpipe have a low number of peptides, which demonstrates that they are present as a result of the increased sensitivity of the DIA-NN/DIA combination.

Reproducibility of the DUBome, visualised as CVs (%) in Figure 2B, shows that both DIA/DIA-NN and DDA/Fragpipe average low CVs of less than 10%. Here, DIA-NN has slightly higher CVs overall when compared to Fragpipe. DIA-NN with and without MBR has some extreme outliers (CV > 40%). The 2 ion minimum quantification for Fragpipe results is the most reproducible setting with CVs < 20%, but at the cost of a reduced DUBome depth.

For the ABPP-HT approach, missing values present a considerable issue, as the absence of a DUB at a high inhibitor concentration may be indicative of complete inhibition, or of a missing value. This issue can be overcome by increasing the number of replicates. However, more replicates mean a lower throughput. This issue has been outlined previously in the context of stochastic DDA IP data [27], and so was considered here for both DIA-NN and Fragpipe analyses. Missing values are extremely low for both DIA-NN and Fragpipe, thereby not presenting an issue for this dataset (Figure 2C). Fragpipe and DIA-NN with MBR had no missing values at all, which is expected given that the MBR feature minimises missing values across runs. The DUBs that did have one or two missing values for both Fragpipe and DIA-NN were quantified from two or less peptides. Inhibition of DUBs with low intensities that have been quantified from two or less peptides should be carefully considered in any case.

### 2.3. Concentration-Dependent Quantification Sensitivity, Precision, and Accuracy with DDA and DIA

To further test the sensitivity of DIA/DIA-NN and DDA/Fragpipe, we performed a titration of peptide injections on the timsTOF after immunoprecipitation of 100 µg of lysate. Without MBR, both DIA-NN and Fragpipe were able to detect a representative panel of 10–15 DUBs with only 50 ng of peptides injected. For both the number of DUBs identified, and the number of proteins detected overall, DIA data with DIA-NN MBR was the most sensitive analysis across the titration (Figure 3A,B).

With confidence in the DUBome depth, reproducibility and sensitivity offered by both DIA/DIA-NN (MBR), and DDA/Fragpipe (MBR 2 ion minimum quant), we applied both workflows to specific and pan-DUB inhibitors to check that the quantitation of the methods agree with each other in the context of DUB inhibition. Inhibition can be quantified by taking the intensity of a DUB immunoprecipitated by the HA-Ub-PA probe in the presence of an inhibitor and normalising it to a control with no inhibitor present. A concentration dependence, for USP7, with the pan-DUB inhibitor PR619 resulted in small differences in quantitation between DDA/Fragpipe quantitation and DIA/DIA-NN quantitation (Figure 3C). Both data sets stray from the expected fit with the same trend (Figure 3D), which may be indicative of the complex kinetics associated with a pan-DUB inhibitor and subsequent HA-Ub-PA binding. Complementing this data, a concentration dependence with the USP7 specific inhibitor FT827 resulted in good agreement between the DDA/Fragpipe and the DIA/DIA-NN methodology (Appendix A).

Quantitation accuracy was further validated by evaluating the difference in the remaining activity between DDA/Fragpipe data and DIA/DIA-NN data for DUBs that were common to both datasets for inhibitors PR619 (Figure 3E) and FT827 (Appendix A). Although there is variability between the Fragpipe and DIA-NN datasets, the variability does not occur in a concentration-dependent manner, meaning that the selectivity of FT827, and the pan-reactivity of PR619, aligns across the two datasets.

### 2.4. Applicability of ABPP-HT* to DUB Inhibitor Screening

FT827 is a highly selective USP7 inhibitor, HBX41108 is a cross-reactive USP7 inhibitor, P22077 is a selective USP7 inhibitor with known cross-reactivity to USP47, and PR619 is a pan-DUB inhibitor [12,17,28]. As a proof-of-concept experiment, inhibitor profiles were assessed across the identified DUBomes by DDA/Fragpipe and DIA/DIA-NN. Four DUBs at the low end of the LC/MS-MS dynamic range were discarded from the DIA/DIA-NN dataset as they did not show a concentration-dependent profile, most likely due to inaccurate quantification at lower abundance. Despite this, DIA/DIA-NN continued to provide an increased sensitivity, with the identification of 29 DUBs, compared to 22 DUBs identified by Fragpipe (Figure 4A).

Both DIA/DIA-NN and DDA/Fragpipe confirmed the previously described inhibitor profiles in their quantitation across the DUBome (Figure 4A). One exception is that Fragpipe does not confirm USP47 inhibition in response to P22077 treatment, as previously described [12,29].

Although missing values were found to be comparable between DIA/DIA-NN and DDA/Fragpipe with probe-labelled samples in triplicate (Figure 2C), when applied across a larger sample cohort, DIA/DIA-NN displays fewer missing values compared to DDA/Fragpipe (Figure 4A). This is especially apparent at high concentrations of PR619 treatment across the DUBome. The quantitation of these values allows for the conclusion that a value is reduced as a consequence of inhibitor treatment, rather than it being missing perhaps as a consequence of inhibitor treatment, or perhaps as a consequence of missing values occurring due to the stochastic nature of DDA LC-MS/MS.

IC_50_ values for all the tested USP7 DUB inhibitors were calculated (Figure 4B). It is of note that HBX41108 and P22077 were also assessed for quantitation overlap, although neither inhibitor is potent enough to lead to accurate IC_50_ values, resulting in large 95% confidence intervals. For FT827, HBX41108 and P22077, the 95% confidence intervals overlap for Fragpipe, DIA-NN and previously published Maxquant data (Figure 4B). For PR619, the USP7 IC_50_ value and 95% confidence intervals align between DIA-NN and Maxquant, but do not overlap with Fragpipe. This combined with the overall smaller 95% confidence intervals and higher R squared values of DIA-NN compared to Fragpipe suggests that the quantitation from the DIA/DIA-NN combination may be more accurate.

## 3. Discussion

The potency and selectivity of an inhibitor in the cellular environment can be highly affected by limited permeability, degradation, and reactivity with off-target events. When a suitable activity-based probe is available, ABPP assays can inform on how the inhibitor is engaging its target in a cellular matrix.

One of the classical limitations of the ABPP assay was its intrinsic low throughput due to probe affinity purifications being performed by hand. This is particularly prevalent when applying this assay for small molecule inhibitor screening. We have recently developed a high-throughput compatible activity-based protein profiling (ABPP-HT) method that allows the semi-automated analysis of multiple samples in a microplate format. To achieve this, we automated the affinity purification and proteomic sample preparation steps via a liquid handling robot combined with the high-throughout-compatible LC-MS/MS platform Evosep/tims TOF [17].

While ABPP-HT increased the throughput of the traditional ABPP approximately ten times, the proteomic depth was significantly reduced. Around 15–25 DUBs were identified with this methodology, clearly below the numbers of the classical ABPP (~30–40 DUBs IDs; [16]) and significantly lower than our previously reported high-depth ABPP (>70 DUB IDs; [13]).

To improve the sensitivity and proteomic depth covered by the ABPP-HT workflow, we decided to test DIA-NN and maxDIA for data-independent acquisition method mass-spectrometry (DIA) [20,22]. DIA allows the quantitation of low-abundance peptides and overcomes the intrinsic missing values limitation of DDA, due to the stochastic nature of top-N fragmentation [30]. Recently, improved DIA search engines have been developed, resulting in substantially improved coverage in complex proteomics phosphoproteomics [31] and ubiquitomics workflows [32,33]. We also compared Fragpipe, a more recently developed proteomics search engine for data-dependent acquisition mass-spectrometry (DDA), against the existing DDA software MaxQuant [23,24,25,26].

When using DIA/DIA-NN, the number of identified DUBs went up to 38 DUBs (matching between runs) in a comparable batch of samples. This is a considerable improvement (~50%) over using MaxQuant combined with DDA. When implementing the DDA search engine Fragpipe we were able to consistently identify up to 28 DUBs. We also tested MaxDIA, the software platform for analysing library-free DIA data using the MaxQuant environment [22]. For ABPP-HT, both MaxQuant and MaxDIA showed less coverage as compared to DIA-NN and Fragpipe so we decided to base our comparisons on these two search engines, reflecting DDA- and DIA-based analysis pipelines.

In terms of reproducibility and repeatability, both Fragpipe and DIA-NN performed comparably when the number of samples was low, as shown by the low coefficients of variation and the near absence of missing values, with or without MBR. However, DIA-NN MBR was consistently more sensitive than Fragpipe when injecting increasing amounts of peptides coming from 100 µg of protein starting material.

Finally, we profiled a panel of USP7 small-molecule inhibitors P22077 [12], HBX41108 [34] and FT827 [28] and pan-DUB inhibitor PR619 [12] using our ABPP-HT* (enhanced high-throughput-compatible activity-based protein profiling). Both search engines managed to provide comparable dose-dependent inhibition profiles for USP7 in lysates treated with a range of inhibitor concentrations. Selectivity profiles for all inhibitors led to expected DUBome reactivity, with FT827 displaying high specificity for USP7, and PR619 displaying high levels of reactivity across a representative panel of 22 (Fragpipe) and 29 (DIA-NN) DUBs, respectively. It is important to note, that when analysing a larger number of samples, DIA-NN not only displays more sensitivity but also a significant reduction in the number of missing values when compared to Fragpipe/DDA. This is highly relevant for this application as it can lead to false positives, and therefore, flawed inhibition and cross-reactivity profiles.

In summary, by optimising the data acquisition and the search engine we managed to increase the proteomic and DUBome depth of our current high-throughput methodology by a ~50%, allowing for fast and deep profiling of representative cellular cysteine peptidase DUBs (Figure 2). As demonstrated here for DUB inhibitors, *ABBP-HT is particularly interesting for drug discovery efforts as it allows for the acquisition of target engagement data (i.e., potency and selectivity) in a cellular context. Another potential application is the DUB profiling of a relatively large cohort sample, suitable for clinical proteomics studies. Finally, this workflow should be compatible with other ABPs engineered to profile other classes of enzymes.

## 4. Materials and Methods

### 4.1. Cell Culture and Lysis

MCF-7 cells (ATCC HTB-22) were cultured and lysed as previously described [17]. Briefly, cells were cultured in high glucose Dulbecco’s Modified Eagle’s Medium (DMEM) with 10% Fetal Bovine Serum, at 37 °C, 5% CO_2_. Cells were washed and scraped in phosphate-buffered saline (PBS), and collected at 300× *g*. Cells were lysed in 50 mM Tris Base, 5 mM MgCl 2·6 H2O, 0.5 mM EDTA, 250 mM Sucrose and 1 mM Dithiothreitol (DTT) (pH 7.5). Lysis was carried out through 10 × 30 s vortexing of the lysate using an equal volume of acid-washed glass beads, with 2 min breaks on ice. Lysates were clarified through centrifugation at 14,000× *g*, at 4 °C for 25 min.

### 4.2. HA-Ub-PA Synthesis

HA-Ub-PA was synthesised as previously described [13,17,35]. HA-Ubiquitin (Gly76del)-intein-chitin binding domain (CBD) was expressed in *E. coli*. The cell pellet was suspended in 50 mM Hepes, 150 mM NaCL, 0.5 mM DTT and sonicated for 30 s × 10, with 30 s breaks. The lysate was purified using Chitin bead slurry, and incubated with 100 mM MesNa overnight at 37 °C to form HA-Ub-MesNa. Incubation with 250 mM propargylamine (PA) at room temperature for 20 min, followed by PD-10 desalting resulted in HA-Ub-PA formation, as confirmed by Western blot and intact protein LC-MS (data not shown).

### 4.3. HA-Ub-PA and Inhibitor Labelling

Lysates were diluted to a 3.33 mg/mL protein concentration, with 250 µg of protein per reaction unless otherwise stated (accounting for dilution with HA-Ub-PA and inhibitor). Lysate was incubated with inhibitors (or Dimethyl sulfoxide (DMSO) for the control) for 1 h at 37 °C. HA-Ub-PA was then incubated with lysate with/without inhibitor labelling for 45 min at 37 °C. Reactions were quenched using NP40 (0.5% *v/v*) and sodium dodecyl sulfate (SDS) (0.5% *w/v*), and diluted to 1 mg/mL lysate protein concentration using NP40 buffer (50 mM Tris, 0.5 % NP40 (*v/v*), 150 mM NaCL, 20 mM MgCl_2,_ pH 7.4).

### 4.4. Agilient Bravo Assay MAP Liquid Handling Platform Immunoprecipitation

The immunoprecipitation methodology used in this paper is as outlined previously using the original ABPP-HT methodology [17]. Briefly, 100 µg of anti-HA antibody (12CA5) was immobilised on Protein A cartridges (Agilent, Santa Clara, CA, USA, G5496-60000), using the in-built immobilisation methodology, with PBS used for all wash steps. The in-built affinity purification methodology was used for HA-Ub-PA immunoprecipitation, with standard settings other than a slow flow-rate for lysate loading (1 µL/min) to ensure optimal antibody binding. Peptides were eluted with 50 µL of 0.15% TFA.

### 4.5. Mass Spectrometry Sample Preparation

Samples were neutralised with 180 µL of 100 mM Triethylammonium bicarbonate (TEAB), pH 8.5, and digested overnight at 37 °C with 1 µg of trypsin (Worthington, Columbus, OH, USA, LS003740 TPCK-treated). Samples were then acidified with formic acid (1% final concentration).

### 4.6. Evosep/timsTOF LC-MS/MS

Solvent A was 0.1% formic acid in water and Solvent B was 0.1% formic acid in acetonitrile. All centrifugation steps were 700× *g* for 60 s unless stated otherwise. For peptide loading onto EvoTips (Evosep, Odense, Denmark) [36], tips were first activated by soaking in 1-propanol, then washed with 20 μL of Solvent B by centrifugation. Washed tips were conditioned by soaking in 1-propanol until the C18 material appeared pale white. Conditioned tips were equilibrated bBruker, Billerica, MA, USAy centrifugation with 20 μL Solvent A. Samples were then loaded into the tips while the tips were soaking in Solvent A to prevent drying, peptides were then bound to the C18 material by centrifugation. Tips were washed by centrifuging with 20 μL Solvent A. Next, 100 μL Solvent A was added to the tips and the tips were centrifuged at 700× *g* for 10 s. Samples were then immediately analysed by LC-MS/MS.

Peptides were analysed using an Evosep One (Evosep) [36] coupled to a timsTOF Pro mass spectrometer (Bruker, Billerica, MA, USA) using a 100 µm × 80 mm C18 column packed with 3 µm beads (PepSep, Marslev, Denmark, EV-1109). The pre-set “100 samples per day” method was used, resulting in a gradient length of 11.5 min at a flow rate of 1.5 µL/min. The timsTOF Pro was operated in parallel accumulation, serial fragmentation (PASEF) mode. TIMS ion accumulation and ramp times were set to 100 ms and mass spectra were recorded from 100–1700 *m*/*z*.

### 4.7. Data-Dependent Acquisition Methods

The ion mobility range was set to 0.85–1.30 Vs/cm^2^. Precursor ions selected for fragmentation were isolated with an ion mobility-dependent collision energy that increased linearly from 27–45 eV over the ion mobility range. Three PASEF MS/MS scans were collected per full TIMS-MS scan, giving a duty cycle of 0.53 s. Ions were included in the PASEF MS/MS scans if they met an intensity threshold of 2000 and were sampled multiple times until a summed target intensity of 10,000; once sampled, ions were excluded from reanalysis for 24 s.

### 4.8. Data-Independent Acquisition Methods

The mass spectrometer was operated in diaPASEF mode using 8 diaPASEF scans per TIMS-MS scan, giving a duty cycle of 0.96 s [19].

For Figure 1C–E and Figure 2C–E and Figure 3C–E and Figure 4 the ion mobility range was set to 0.6–1.6 Vs/cm^2^. Each mass window isolated was 25 m/z wide, ranging from 400–1000 *m*/*z* with an ion mobility-dependent collision energy that increased linearly from 20 eV to 59 eV between 0.6–1.6 Vs/cm^2^ (Appendix A).

Due to a software upgrade, for Figure 3A,B the ion mobility range was set to 0.85–1.3 Vs/cm^2^. Each mass window isolated was 25 m/z wide, ranging from 475–1000 *m*/*z* with an ion mobility-dependent collision energy that increased linearly from 27 eV to 45 eV between 0.85–1.3 Vs/cm^2^ (Appendix A).

### 4.9. Software Settings

All software was set to default settings unless stated (e.g., MBR vs. no MBR), with N-terminal acetylation and Methionine oxidation set as variable modification, and no fixed modifications. All searches used *Homo sapiens* Uniprot database (retrieved 16 April 2021), other than maxDIA which requires its own generated FASTA (UP000005640_9606). Software versions: Fragpipe 17.1 (MSFragger 3.4, Philosopher 4.1.1, Python 3.9.7). DIA-NN 1.8, Maxquant 2.0.3.

### 4.10. Data Analysis

Graphs were generated and fitted using Graphpad prism 9.2.0 (333), other than the upset plot (Figure 1B) [37]. For Fragpipe/Maxquant/maxDIA unique/razor ‘maxLFQ’ intensities were used, for DIA-NN razor intensities are not assigned to a protein group and so unique intensities were used in Figure 1, Figure 2 and Figure 3 to avoid overestimating the number of DUBs present, and in Figure 3A,B to avoid counting the same protein multiple times. Figure 1D and Figure 4 includes DIA-NN razor intensities, which are denoted as DUB1; DUB2. Unique Fragpipe peptides were counted from the output file “protein.tsv”. Proteotypic DIA-NN peptides with intensities > 0 were extracted from the output file report.pr_matrix.tsv, with precursors averaged to give unique peptide numbers.

## Data Availability

The mass spectrometry proteomics data have been deposited in the ProteomeXchange Consortium via the PRIDE [38] partner repository with the dataset identifier PXD031848.

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
