# Peer review of "ABPP-HT*—Deep Meets Fast for Activity-Based Profiling of Deubiquitylating Enzymes Using Advanced DIA Mass Spectrometry Methods"

_ijms, 2022, doi:10.3390/ijms23063263_

Round 1

Reviewer 1 Report

In this paper, the authors compared different LC-MS/MS methods including different software tools to improve the activity-based profiling of deubiquitylating enzymes. The result is interesting and valuable, I recommend this work for publication.

Author Response

We would like to thank the reviewer for the positive feedback.

Reviewer 2 Report

This manuscript contains too many abbreviations of proper nouns, it is difficult for readers to understand the article.

Comments:

  1. List a Table to explain the abbreviations of these proper nouns.
  2. Please also give the suitable definitions of the abbreviations shown in the Figures.
  3. The word "activitomics" is redundant.
  4. The word "chemical biology" is redundant.
  5. Line 366 to 372, why  cellular cysteine peptidase DUBs  are associated with drug discovery?

Author Response

We would like to thank this reviewer for his feedback.

This manuscript contains too many abbreviations of proper nouns, it is difficult for readers to understand the article. Apologies for this. We have now written the "proper" complete noun plus the abbreviation the first time they are mentioned and we have also created an abbreviations section in the manuscript to help readers to follow the science in the text.

Comments:

  1. List a Table to explain the abbreviations of these proper nouns. We have created an abbreviations section in the text (Page 14; lines 481-490).
  2. Please also give the suitable definitions of the abbreviations shown in the Figures. This has been now been clarified also in the matching figure legends.
  3. The word "activitomics" is redundant. Activitomics is just a keyword and it means "techniques that provide a measure of enzyme and pathway activation".  We believe this is a relevant keyword for this article.
  4. The word "chemical biology" is redundant. The same applies to chemical biology (the study of the chemicals and chemical reactions involved in biological processes). We think the ABPP assay is a clear application of chemical biology.
  5. Line 366 to 372, why  cellular cysteine peptidase DUBs  are associated with drug discovery? We used an activity-based probe against this class of enzymes in this study but this does not mean this workflow cannot be used for additional enzyme families. This has been now clarified in the text (Page 12; lines 380-381). Apologies for this misunderstanding, we did not mean that DUBs are associated with drug discovery.

Reviewer 3 Report

In the present manuscript, Jones H.B.L., et al. report on the extension of their well established, activity-based protein profiling platform for DUB inhibitors by mass-spectrometry. Integrating recent instrumental and computational advances in the field of MS-based proteomics, the authors provide an impressive showcase for the application of such technologies in the development of DIA (rather than DDA)-based high-throughput assays in drug-discovery workflows.The ultimate, optimized workflow allowed for the consistent and confident profiling of 35-40 DUBs whilst maintaining the considerable throughput of 100 samples per day. The paper is very well and clearly written, scientifically sound, and of interest to field. I recommend publication of the manuscript with only minor changes (please see below).

Minor Comments:

line 58 - "These 74 DUBS represented"; should be DUBs

line 144 - "It is worth noting that for 1 ion minimum quantitation, with or without MBR, samples, all identified DUBs had a CV of <=20%, suggesting that the increased depth attained with this reduced stringency is not negatively impacting the quantitative accuracy of the identified DUBome."; not clear! remove "samples,"?

line 162 "To ascertain whether the additional DUBs identified by both methods are as a di-rect result of HA-Ub-PA enrichment, the HA-Ub-PA samples were compared to a no HA-Ub-PA control sample (Figure 1C and 1D)."; not clear! remove "as"?

line 175 "These additional proteins including the DUBs detected in the no probe sample will be sourced from co-immunprecipitation of protein complexes as well as non-specific background binding of proteins to the antibody and beads during the immunoprecipitation."; "will be sourced"? please re-phrase.

Figure 2A - The current scaling of the x-axis does not allow to read the number of peptide IDs for most proteins. I recommend adjusting the x-axis to e.g. maximum number of theoretical tryptic peptides for each protein, or % sequence coverage.

Figure 3E - Difference plot: this is extremely hard to read! please use unique symbols for the respective inhibitor concentration levels.

line 390 "The cell pellet was suspended in 50 mM Hepes, 150 mM NaCL, 0.5 mM DTT and sonicated for 30 seconds X 10, with 30 second breaks." - should be "x" (minor)?

Scheme 2 - This is a very redundant figure! I suggest removing it.

Figure S1.C - is the "color code" in panel C intended? please either explain its meaning in the figure legend or remove.

Author Response

In the present manuscript, Jones H.B.L., et al. report on the extension of their well established, activity-based protein profiling platform for DUB inhibitors by mass-spectrometry. Integrating recent instrumental and computational advances in the field of MS-based proteomics, the authors provide an impressive showcase for the application of such technologies in the development of DIA (rather than DDA)-based high-throughput assays in drug-discovery workflows.The ultimate, optimized workflow allowed for the consistent and confident profiling of 35-40 DUBs whilst maintaining the considerable throughput of 100 samples per day. The paper is very well and clearly written, scientifically sound, and of interest to field. I recommend publication of the manuscript with only minor changes (please see below).

We would like to thank the reviewer for the positive and constructive feedback.

Minor Comments:

line 58 - "These 74 DUBS represented"; should be DUBs. Thank you, this has now been amended.

line 144 - "It is worth noting that for 1 ion minimum quantitation, with or without MBR, samples, all identified DUBs had a CV of <=20%, suggesting that the increased depth attained with this reduced stringency is not negatively impacting the quantitative accuracy of the identified DUBome."; not clear! remove "samples,"? Thank you, this has now been amended.

line 162 "To ascertain whether the additional DUBs identified by both methods are as a di-rect result of HA-Ub-PA enrichment, the HA-Ub-PA samples were compared to a no HA-Ub-PA control sample (Figure 1C and 1D)."; not clear! remove "as"? Thank you, this has now been amended.

line 175 "These additional proteins including the DUBs detected in the no probe sample will be sourced from co-immunprecipitation of protein complexes as well as non-specific background binding of proteins to the antibody and beads during the immunoprecipitation."; "will be sourced"? please re-phrase. Thank you, this has now been amended.

Figure 2A - The current scaling of the x-axis does not allow to read the number of peptide IDs for most proteins. I recommend adjusting the x-axis to e.g. maximum number of theoretical tryptic peptides for each protein, or % sequence coverage. Apologies for this. The reviewer is right and we have now ranked the DUBs by abundance and amended the x-axis scaling so the number of peptides is now visible for all the proteins.

Figure 3E - Difference plot: this is extremely hard to read! please use unique symbols for the respective inhibitor concentration levels. Thank you, this has now been amended.

line 390 "The cell pellet was suspended in 50 mM Hepes, 150 mM NaCL, 0.5 mM DTT and sonicated for 30 seconds X 10, with 30 second breaks." - should be "x" (minor)? Thank you, this has now been amended.

Scheme 2 - This is a very redundant figure! I suggest removing it. We thank the reviewer for this suggestion, although we still believe this scheme provides a nice overview of the existing ABPP assays for the profiling of DUBs and it should help new users to identify the right assay.

Figure S1.C - is the "color code" in panel C intended? please either explain its meaning in the figure legend or remove. Apologies for this, the colour red was used to indicate outliers coming from missing values that could lead to the wrong impression of compound inhibition. This is explained in the matching figure legend.